# Photonic Crystal Enhanced by Metamaterial for Measuring Electric Permittivity in GHz Range

**Arafa H. Aly [1],\*, Ayman A. Ameen [1,2], M. A. Mahmoud [2], Z. S. Matar [3], M. Al-Dossari [4] and Hussein A. Elsayed [1]**

[1]    TH-PPM Group, Physics Department, Faculty of Science, Beni-Suef University, Beni Suef 62511, Egypt
[2]    Physics Department, Faculty of Science, Sohag University, Sohag 82749, Egypt
[3]    Department of Physics, Faculty of Applied Science, Umm-Al-Qura University, Mecca 24381, Saudi Arabia
[4]    Physics Department, Dahran Aljanoub, King Khalid University, Abha 61421, Saudi Arabia
\*    Correspondence: arafa.hussien@science.bsu.edu.eg

**Abstract:** The rise of broadband cellular networks and 5G networks enable new rates of data transfer. This paper introduces a new design to measure the permittivity in the GHz range of non-magnetic materials. We tested the proposed design with a wide range of materials such as wood, glass, dry concrete, and limestone. The newly proposed design structure has a maximum sensitivity of 0.496 GHz/RIU. Moreover, it can measure permittivities in the range from 1 up to 9. The main component of the designed structure is a defective one-dimensional photonic crystal with a unit cell consisting of metamaterial and silicon. In addition, we demonstrate the role of the metamaterial in enhancing the proposed design and examine the impact of the defect layer thickness on the proposed structure.

**Keywords:** photonic crystals; metmaterials; permittivity; sensitivity





## 1. Introduction

New advances in material science have enabled breakthroughs in technologies and applications. One of these innovations is the ability to control and manipulate the propagation of photons using artificial structures. One of the most researched photonic engineered structures is photonic crystals (PCs). Photonic bandgap materials, or photonic crystals, are periodic structures whose refractive index periodicity functions [1,2]. The range of frequencies is prohibited from propagation in these artificial structures due to Bragg scattering; this range is called photonic bandgap (PBG) [3].

In 1987, Yablonovitch first introduced PCs in their three-dimensional form [4,5]. After that, various types of photonic crystals were researched. As a result, PCs can be classified in several ways. For example, PCs could be classified through the number of dimensions that refractive index periodicity that extends into one [6,7], or two [8], or three-dimensional photonic crystals [1]. Although three-dimensional PCs are the only type with a complete photonic bandgap, one-dimensional PCs are the most researched type due to the ease of fabrication [9,10].

PCs have been investigated for numerous efforts in various applications in recent years, such as electromagnetic wave filters [11], cavities [12], waveguides [13], and optical communications [14]. By introducing a defect inside the periodic structure, a defect peak appears inside the PBG [15]. By modifying the refractive index of the defect, the properties of the defect peak can be altered [16,17]; such tunability opens the door for many different sensors, extending from chemical compounds to different biological cells and tissues [18–21]. Many materials have been examined to enhance the PCs' photonic bandgap properties and sensing abilities, ranging from metals [22], dielectrics [23,24], semiconductors [17], superconductors [25], and plasma. Recently, metamaterials have been identified as a possible candidate to enhance the PCs' properties [26].

Although metamaterials don't have a formal definition yet, many artificial structures have become prime examples of metamaterials. Photonic metamaterials are one of the most researched subsets of metamaterials. Metamaterials are structures engineered to exhibit properties like the negative refractive index (Left-handed) [27] or epsilon-near-zero (ENZ) [28]. Metamaterials have a lot of prospective usages, such as transformation optics [29], optical hyperlens [30], and invisible cloaking [31]. The properties of metamaterial do not come from their chemical bonding but rather from their engineered structure [32]. A few examples of the studied metamaterial structures are designed and fabricated in the form of non-magnetic split-ring resonators [33,34], short-slab pairs [35], and cascaded fishnets [36].

## 2. Theoretical Analysis

In this section, we demonstrate the main theoretical framework for our simulation procedure. The proposed structure is composed of alternating layers of metamaterial and silicon with a defect layer in the middle of the structure. The design configuration is expressed as $(M_A M_B)^{\frac{N}{2}} M_f (M_B M_A)^{\frac{N}{2}}$, as shown in Figure 1. The metamaterial consists of a metallic fork between two metal split-ring resonators. Such a configuration has unique properties, such as a double negative refractive index in a certain range of frequencies. The metamaterial structure was simulated by Jensen Li et al. using finite difference time domain (FDTD) and effective electric permittivity $\varepsilon_A$ and effective magnetic permeability $\mu_A$ were obtained [37], as the following:

$$(f) = 1 + \frac{5^2}{0.9^2 - f^2} + \frac{10^2}{11.5^2 - f^2} \tag{1}$$

$$\mu(f) = 1 + \frac{3^2}{0.902^2 - f^2} \tag{2}$$

The well-known Transfer Matrix Method (TMM) is used to calculate the transmittance of the proposed structure [24,38]. The TMM describes each layer $j$ by a matrix $M_j$ which can be calculated as:

$$M_j = \begin{bmatrix} \cos(\delta_j) & -\frac{i}{p_j}\sin(\delta_j) \\ -ip_j\sin(\delta_j) & \cos(\delta_j) \end{bmatrix} \tag{3}$$

For TE mode, $\delta_j$, and $p_j$ are given by:

$$\delta_j = \frac{2\pi d_j}{\lambda} n_j \cos(\theta_j), \ p_j = n_j \cos(\theta_j) \tag{4}$$

Here, $M_j$ is the matrix of the layer $j$ with refractive index $n_j$ and thickness $d_j$. The matrix that describes the whole proposed structure is written as:

$$M = (M_A M_B)^{\frac{N}{2}} M_f (M_B M_A)^{\frac{N}{2}} = \begin{bmatrix} M(1,1) & M(1,2) \\ M(2,1) & M(2,2) \end{bmatrix} \tag{5}$$

In the above expression, $M_A$ and $M_B$ are the matrices of the unit cell layers, $M_f$, and $N$ are the matrix of the defect layer and the periodicity number, respectively. Then, the transmission coefficient for the structure matrix $M$ is defined by:

$$t = \frac{2\, p_0}{\left(M(1,1) + M(1,2)p_f\right)p_0 + \left(M(2,1) + M(2,2)p_f\right)} \tag{6}$$

In the transmission coefficient formula, $p_{0,f} = \sqrt{\frac{\varepsilon_0}{\mu_0}} n_{0,f} \cos\left(\theta_{0,f}\right)$ for the initial and the final medium, respectively. Finally, the transmittance is obtained by:

$$T = \left| t^2 \right| \tag{7}$$

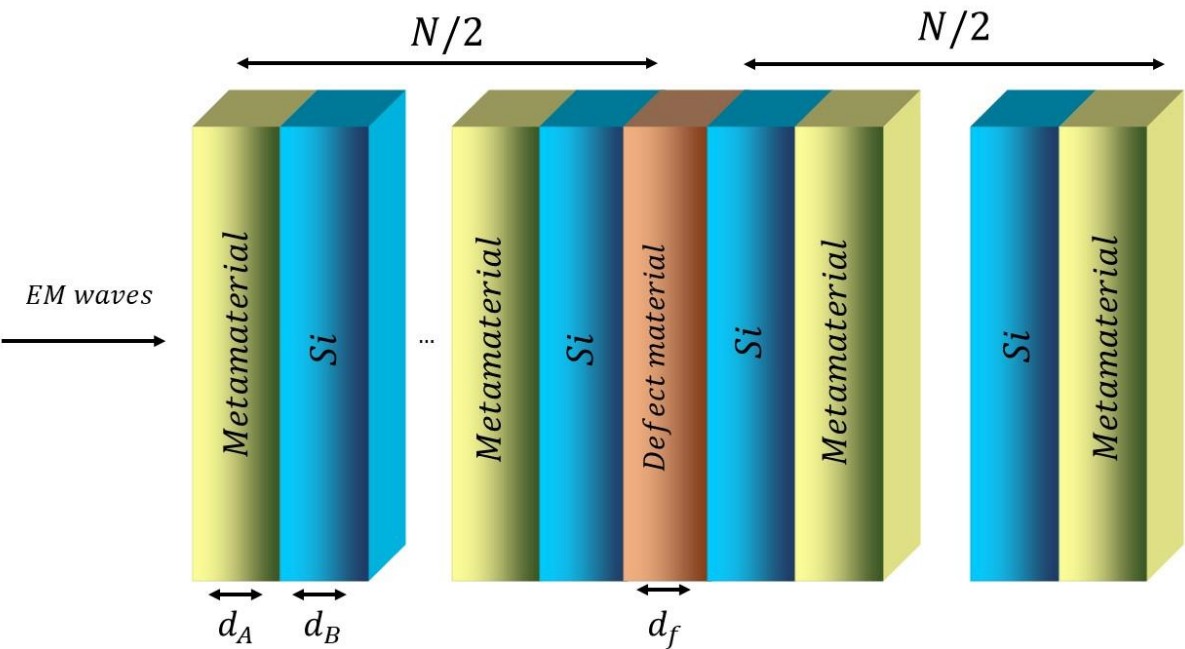

**Figure 1.** Schematic diagram of the proposed photonic metamaterial structure, which consists of bi-layer unit cell repeated by *N* times.

### 3. Results and Discussions

In this section, the simulation results of the proposed device are studied. The primary purpose of the proposed structure is to measure the permittivity of a material. To achieve this goal, we designed a PC structure that consists of a bi-layer unit cell with a defect in the middle of the structure. The material with unknown permittivity is placed in the defect layer. Thus, a defect peak corresponding to its permittivity appears inside the PBG. By determining the exact defect peak parameters, the permittivity of the defect material is calculated. The proposed structure consists of altering layer of metamaterial and silicon with a configuration as $(Metamaterial/Si)^{\frac{N}{2}} \, Defect \, (Si/Metamaterial)^{\frac{N}{2}}$. The metamaterial layer has a thickness $d_A = 31.5$ mm with refractive index $\sqrt{\varepsilon_A \mu_A}$, given by Equations (1) and (2). Meanwhile, the silicon layer has a thickness equal to 9 mm with a refractive index of 3.46. The periodicity number is chosen to be $N = 6$. In this paper, all simulation results are calculated in TE mode for normal incidence $\theta_0 = 0$.

First, the transmittance of the proposed device has been studied, as shown in Figure 2. Here, the defect layer is tested for permittivity from $\varepsilon_f = 4$ up to $\varepsilon = 6$ with a thickness $d_f = 15$ mm. The transmittance shows a PBG between 3.32 GHz and 4.3 GHz with a width equivalent to 0.98 GHz. When the defect permittivity equals $\varepsilon_f = 2$, the defect peak exists at 4.116 GHz, while it shifts to 3.77 GHz, and 3.559 GHz, as the permittivity changes to $\varepsilon_f = 4$, and 6, respectively. Table 1 summarises the parameters of the defect peak as the defect permittivity changes. The quality factor is a dimensionless quantity and can be obtained from the following equations:

$$Q = \frac{f_{peak}}{W_{FWHM}} \tag{8}$$

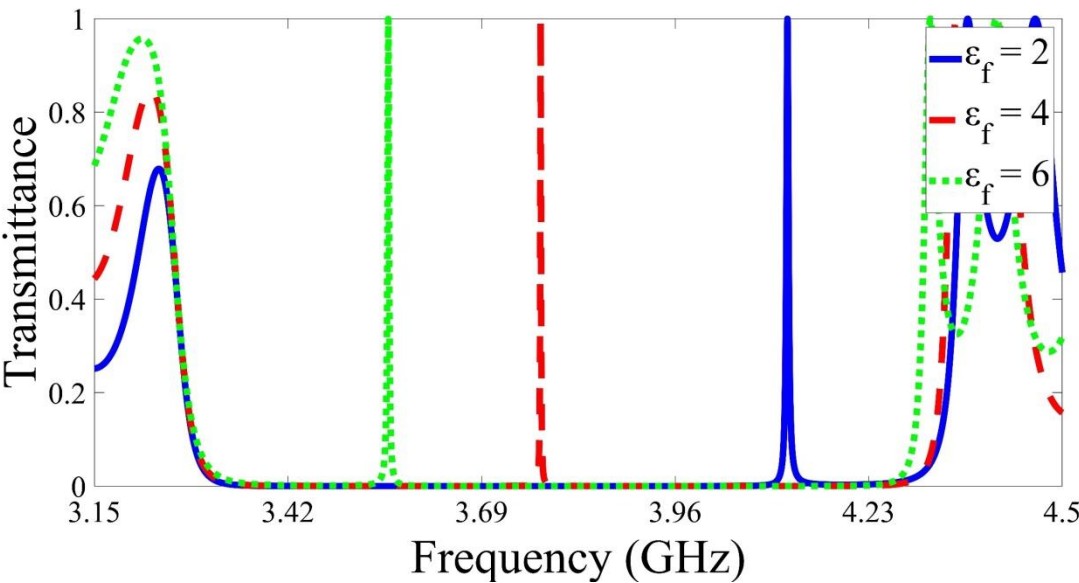

**Figure 2.** The calculated transmittance of the proposed design in the GHz frequency range. The designed structure is consisted of altering layers of metamaterial and silicon with the target material as a defect layer with permittivity equals $\varepsilon_f = 2$, 4, and 6.

**Table 1.** Parameters of the defect peak as $\varepsilon_f = 2$, 4, and 6.

| $\varepsilon_f$ | $f_{peak}$ (GHz) | $W_{FWHM}$ (MHz) | $I_{max}$ | $S$ (GHz/RIU) | $FoM$ (RIU)$^{-1}$ |
|---|---|---|---|---|---|
| 2 | 4.116 | 3.5 | 99.9% | 0.35 | 100.7 |
| 4 | 3.772 | 1.2 | 99.9% | 0.49 | 408.5 |
| 6 | 3.559 | 2.5 | 99.9% | 0.48 | 193.9 |

In the above equation, $f_{peak}$ is the frequency that the defect peak appears, and $W_{FWHM}$ is the full width at half the maximum of the defect peak. Here, $I_{max}$ is the normalized intensity of the defect peak. Additionally, the sensitivity $S$ measures the shift of the defect peak corresponding to the difference in the refractive index and can be obtained from:

$$S = \frac{\Delta f_{peak}}{\Delta n} \tag{9}$$

In the above sensitivity expression, $\Delta f_{peak}$ is the difference between defect peak frequencies, and $\Delta n$ is the difference between refractive indices. Here, the difference is calculated between non-magnetic defect material, which has a refractive index $n_f = \sqrt{\varepsilon_f}$ with a defect peak frequency $f_{peak}$ and air $n_{air} = 1$ which its defect peak occurs at 4.262 GHz. The figure of merit $FoM$ is the sensitivity $S$ of structure over the full width at half maximum $W_{FWHM}$.

$$FoM = \frac{S}{W_{FWHM}} \tag{10}$$

Figure 3 illustrates the influence of changing the permittivity of the defect peak on the transmittance. As the permittivity of the defect material alters from 1 to 9, the position of the main defect peak shifts towards lower frequencies. At low values of the permittivity, the width of the main defect peak starts decreasing with increasing the values of defect permittivity. Then, it begins to increase again with increasing permittivity at high defect layer permittivity values. Moreover, a second defect peak appears inside the photonic bandgap at high values of the defect layer permittivity. The secondary defect peak will not be included in our study.

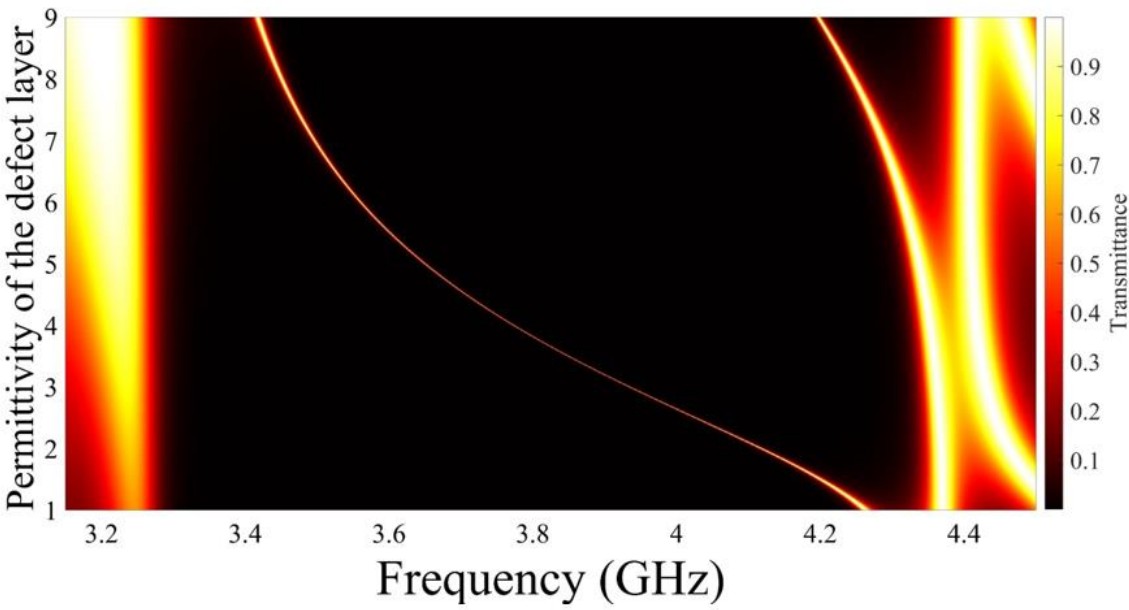

**Figure 3.** The influence of altering the defect layer permittivity on the transmittance.

Figure 4 presents a more quantitive analysis of the peak properties. The defect peak frequency decreases with raising the defect layer permittivity, as shown in Figure 4. The full width at half maximum is shrinking with rasing defect permittivity until it reaches 1.22 MHz at defect layer permittivity equals 3.69; then, it starts raising again as the value of defect layer permittivity increases. For example, when the defect permittivity equals 2, the defect peak appears at 4.116 GHz with a full width at half maximum equivalent to 3.579 MHz. Meanwhile, it reaches 3.417 GHz with a full width at half maximum equals 8.529 MHz as the defect layer permittivity equivalent to 9. The frequency of the defect peak $f_{peak}$ in GHz can be fitted with the defect layer permittivity $\varepsilon_f$ as the following:

$$\varepsilon_f = 58.3174\, f_{peak}^4 - 903.267\, f_{peak}^3 + 5247.1582\, f_{peak}^2 - 13555.37\, f_{peak} + 13149.38 \qquad (11)$$

Thus, through the previous equation, the permittivity of a material can be measured through the proposed structure. The unknown permittivity material can be placed in the defect layer position; a defect peak will appear inside the PBG. Then, the permittivity of the material can be determined by substituting with the defect peak frequency into the previous equation.

After that, the sensitivity and the figure of merit of the proposed design have been studied, as shown in Figure 5. The sensitivity of the proposed structure improves with rising the defect permittivity until it peaked at defect permittivity $\varepsilon_f = 4.72$, which corresponds to the maximum sensitivity of 0.496 GHz/RIU. Then, the sensitivity of the device drops with a further increase in defect permittivity. Similarly, the figure of merit of the proposed device has similar behavior. The figure of merit grows with increasing the defect permittivity; then it reaches its highest value of 397.8 $(\text{RIU})^{-1}$ at defect permittivity $\varepsilon_f = 3.96$. With a further increase of the defect layer permittivity, the figure of merit starts to decrease. After that, the figure of merit continues to decrease as the defect permittivity increases.

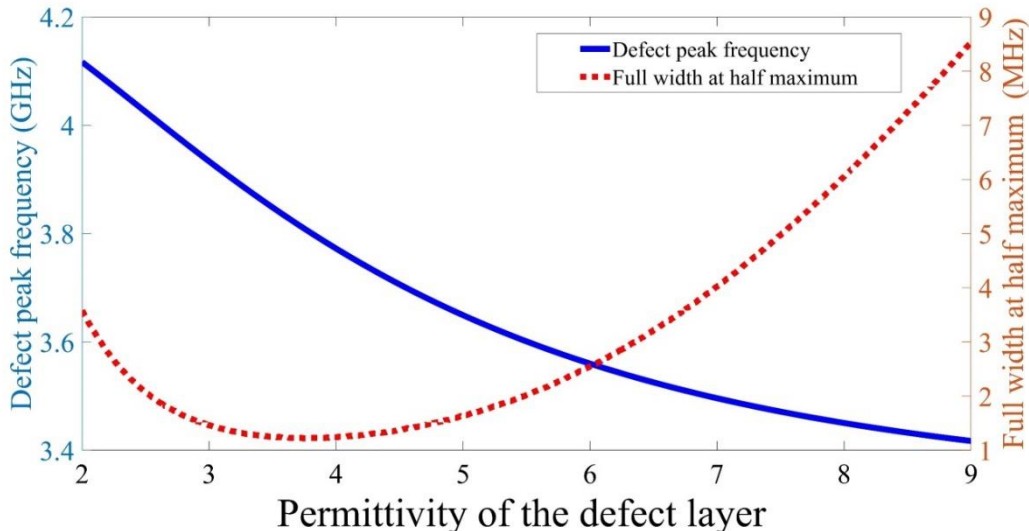

**Figure 4.** The effect of the defect layer permittivity on both the defect peak frequency (solid blue curve) and full width at half maximum of the defect peak (red dotted curve).

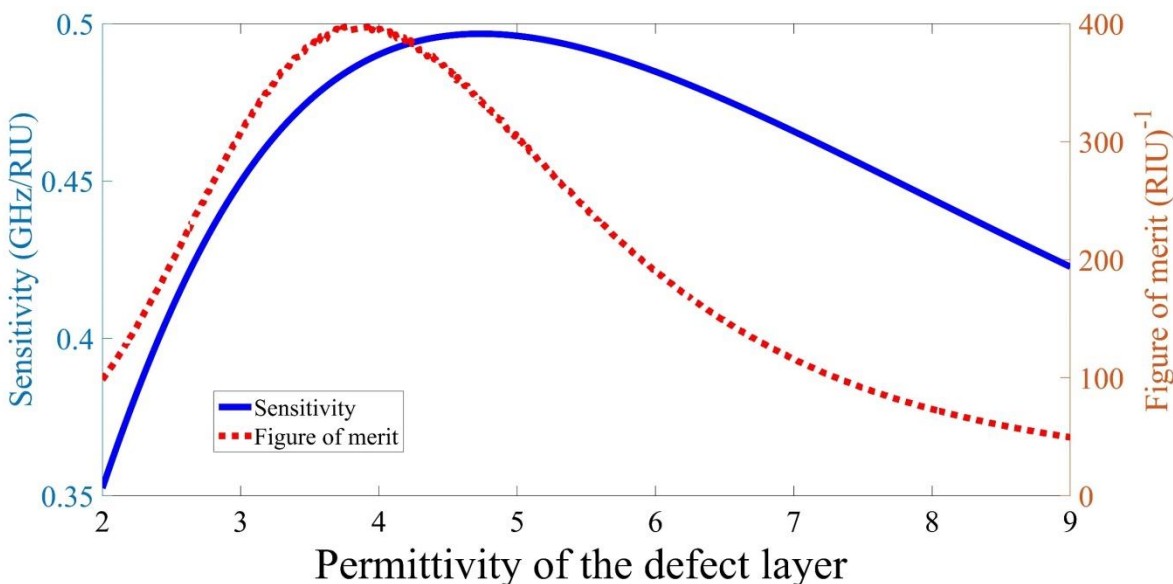

**Figure 5.** The sensitivity and the figure of merit of the proposed structure as the defect permittivity increases.

Then, we analyzed the proposed device for everyday materials, as shown in Figure 6. The refractive indices of the selected materials are listed in Table 2 [39]. For materials with permittivity between 3 and 6, such as dry concrete and glass, the defect peaks for these materials appear around the center of the PBG with relatively high sensitivity and figure of merit, as listed in Table 2. On the other hand, the defect peaks of the materials with permittivity greater than 6 or less than 2 appear in the edges of the PBG. For example, the defect peaks of wood and limestone appear at 4.19 GHz, and 3.47 GHz, respectively. Moreover, these materials' sensitivity is relatively low compared to materials such as dry concrete or glass.

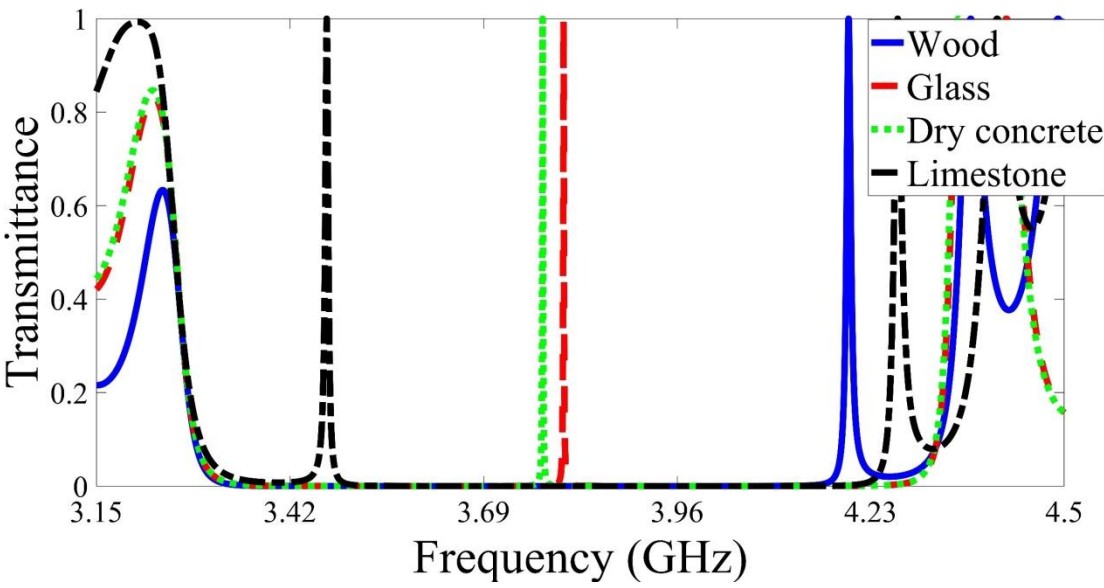

**Figure 6.** Examining the proposed device for different everyday objects.

**Table 2.** Defect peak parameters for the selected everyday materials with their corresponding permittivity.

| Material | $\varepsilon_r$ | $f_{peak}$ (GHz) | $W_{FWHM}$ (MHz) | $I_{max}$ | $S$ (GHz/RIU) | $FoM$ (RIU)$^{-1}$ |
|---|---|---|---|---|---|---|
| Wood | 1.5 | 4.19 | 6.9 | 99.9% | 0.282 | 40.9 |
| Glass | 3.8 | 3.80 | 1.2 | 99.9% | 0.485 | 404.6 |
| Dry concrete | 4 | 3.77 | 1.2 | 99.9% | 0.490 | 408.5 |
| Limestone | 7.5 | 3.47 | 4.9 | 99.9% | 0.455 | 92.8 |

　　　To prove the importance of using metamaterials, we have replaced the metamaterial with glass, as shown in Figure 7. In this simulation, the parameters of the photonic crystals are fixed as the initial parameters, while the configuration of the structure is changed to be $(Glass/Si)^{\frac{N}{2}}$ $Defect$ $(Si/Glass)^{\frac{N}{2}}$ with defect layer permittivity equals $\varepsilon_f = 4$. The thickness of the glass layer is taken to be 25 mm. The advantage of the structure with metamaterial over the structure with glass is very clear, as shown in Figure 7. The full width at half maximum of the defect peak drops from 86 MHz to just 1.2 MHz when the metamaterial is used. Thus, the quality factor Q is increased dramatically using metamaterial. Table 3 summarizes the position $f_{peak}$, full width at half maximum $W_{FWHM}$, intensity $I_{max}$, and the quality factor Q of defect peak for the two structures. Table 3 also shows the advantage of using metamaterial. For example, the periodicity number of the structure with the glass needs to be increased to 16 to achieve similar results as the structure with metamaterial.

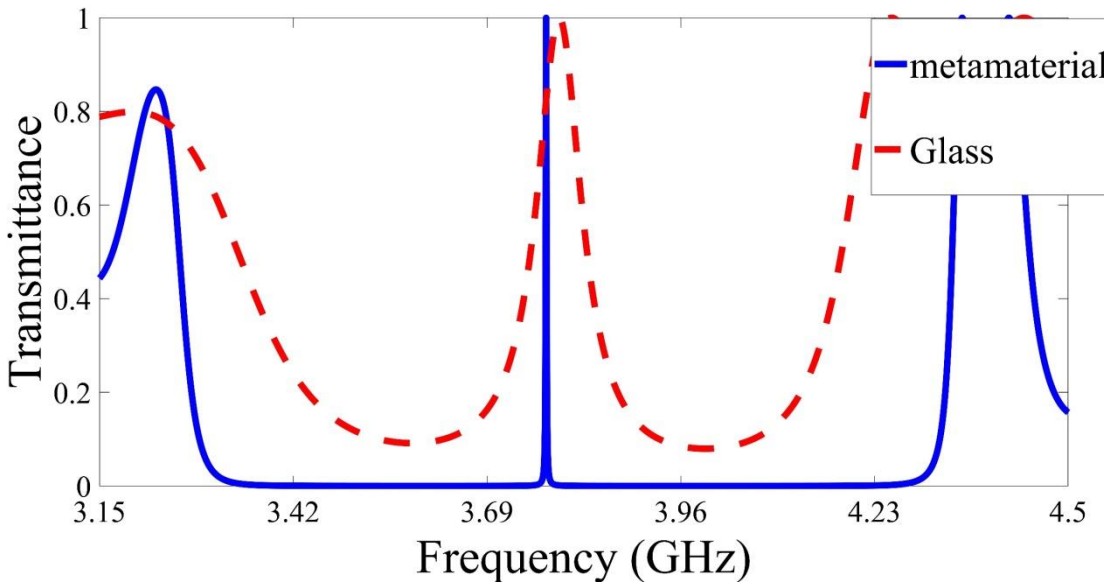

**Figure 7.** Transimmtace of the proposed structure with a unit cell composed of metamaterial/silicon and glass/silicon.

**Table 3.** The effect of replacing the metamaterial with Glass inside photonic crystals on the defect peak parameters.

| Material | N | $f_{peak}$ (GHz) | $W_{FWHM}$ (MHz) | $I_{max}$ | Q |
|---|---|---|---|---|---|
| Metamaterial | 6 | 3.77 | 1.2 | 99.9% | 3143.6 |
| Glass | 6 | 3.79 | 86 | 99.9% | 44.06 |
| Glass | 16 | 3.77 | 1.6 | 99.9% | 2356.2 |

Figure 8 depicts the variances in the transmittance when changing the defect layer thickness. Here, the photonic crystal configuration is the same as the initial configuration at $(Metamaterial/Si)^{\frac{N}{2}} \, Defect \, (Si/Metamaterial)^{\frac{N}{2}}$. Here, the layer permittivity equals $\varepsilon_f = 4$. The edges of the photonic bandgap do not have significant changes; meanwhile, the defect peak frequency is shifted towards lower frequency when increasing the defect layer thickness from 10 mm up to 20 mm.

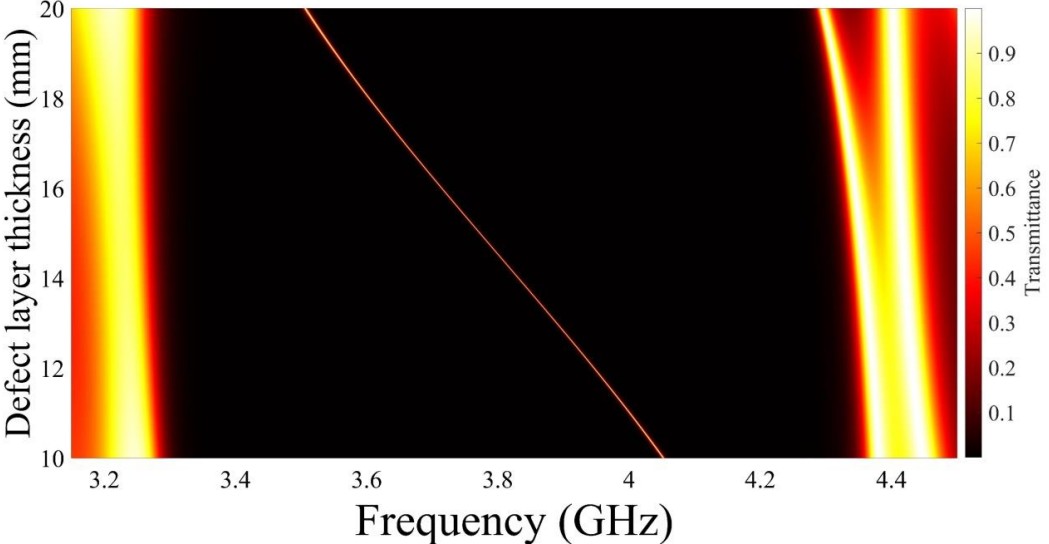

**Figure 8.** The influence of the defect layer thickness on the transmittance of the proposed device.

Finally, the parameters of the defect peak were studied, as shown in Figure 9. With increasing of the defect layer thickness, the defect peak shifted to lower frequencies. Moreover, the defect peak frequency $f_{peak}$ is linearly fitted to the thickness of the defect layer as:

$$f_{peak}(\text{GHz}) = -0.05616 \, d_f \, (\text{mm}) + 4.617 \tag{12}$$

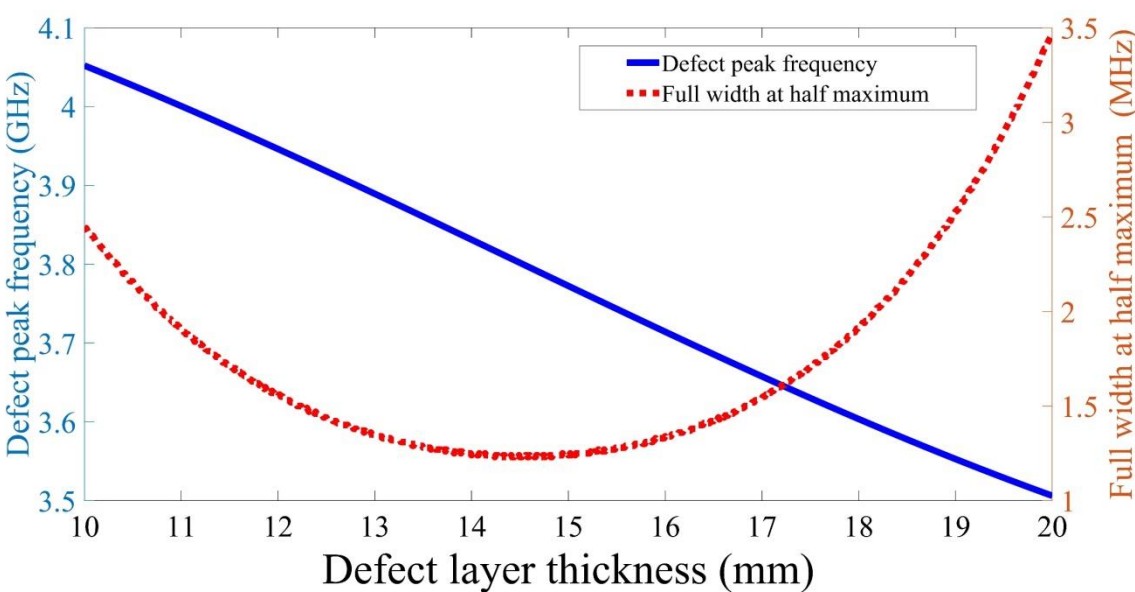

**Figure 9.** The impact of the defect layer thickness on the defect peak frequency and its full width at half maximum.

Besides, the full width at half maximum decreases with the defect layer thickness until it reaches 1.23 MHz at defect thickness equals 14.44 mm; after that, the full width at half maximum increases with further increase of the defect thickness.

## 4. Conclusions

In conclusion, we have introduced a newly designed structure for measuring the permittivity of non-magnetic materials. We have considered such a design for measuring permittivity between $\varepsilon_f = 1$, and $\varepsilon_f = 9$. The proposed structure is designed from a one-dimensional photonic crystal with a defect in the middle. The photonic crystal consists of a metamaterial/silicon unit cell with a periodicity number equal to six. To measure the permittivity of a material, the targeted material is placed in the defect layer position. Thus, the permittivity of the targeted material can be determined through the defect peak position. The maximum sensitivity of the proposed structure is 0.496 GHz/RIU at permittivity $\varepsilon_f = 4.72$, while the maximum figure of merit is investigated at defect layer permittivity $\varepsilon_f = 3.96$. Additionally, the defect peak is shifted to lower frequencies with increasing the defect layer thickness.

**Author Contributions:** Conceptualization, A.H.A. and A.A.A.; methodology, A.A.A.; software, A.A.A.; validation, M.A.M., H.A.E. and Z.S.M.; formal analysis, A.A.A.; investigation, H.A.E.; resources, M.A.-D.; data curation, A.A.A.; writing—original draft preparation, A.A.A. and M.A.M.; writing—review and editing, H.A.E., A.H.A. and M.A.-D.; visualization, A.H.A., A.A.A. and M.A.M.; supervision, A.H.A.; project admin-istration, A.H.A. amd M.A.M.; funding acquisition, A.H.A., M.A.-D. and Z.S.M. All authors have read and agreed to the published version of the manuscript.

**Funding:** This research received no external funding.

**Acknowledgments:** This work was produced with the financial support of the Academy of Scientific Research and Technology of Egypt; ScienceUP/GradeUp initiative: Grant. Agreement No (7859). Its contents are the sole responsibility of the authors and do not necessarily reflect the views of the Academy of Scientific Research and Technology.

**Conflicts of Interest:** The authors declare that there are no conflict of interest.

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
