# Peer review of "Photonic Crystal Enhanced by Metamaterial for Measuring Electric Permittivity in GHz Range"

_photonics, doi:10.3390/photonics8100416_

Round 1

Reviewer 1 Report

The manuscript presents  a new design to measure the in the GHZ range of non-magnetic materials. The paper has listed some example for the trial of the measurements, demonstrating the significance of the new design to the understand of the new optic materials. The manuscript should be published after referring the following issue.

  1. I have one doubt about the data in Table 2, the value is permitivity or refractive index? the experssion is confusion in the manuscript.
  2. Why design such a structure to measure the permittivity of non-magnetic materials? Is this structure easy to construct in practice? Is there a widely used method for measuring the permittivity of non-magnetic materials?
  3. P4: The last paragraph says that as the permittivity of the defect layer increases, another defect peak appears in the high-frequency region, but there are three additional peaks appearing in the higher-frequency region in Figure 3. How to understand?
  4. P3: In addition to the defect peaks, if adding a layer to the original periodic structure will affect the original band gap position of the material? Is there a clear difference from the newly added defect peak position? Will it affect the judgment of permittivity?
  5. Can the permittivity test range in this paper be adjusted by adjusting the refractive index or thickness of the photonic crystal structure? Are there any simulation results in this regard
  6. Will the transparency of the defect layer itself affect the position of the defect peak and thus affect the measurement of permittivity?

Author Response

Dear Prof 
Thanks a lot for your valuable comments We have considered all your comments in the attached file

Regards

Authors

Reviewer 2 Report

In this paper, measurement method of permittivity utilizing one-dimensional photonic crystal with meta-material is proposed. It is shown that the permittivity of material with thickness of about 15mm. I have some questions.

1) In this system, the peak frequency is sensitive to not only permittivity but also defect layer thickness. Therefore, in order to measure permittivity with high accuracy, material thickness seems to be required to be accurately measured. By the way, the FWHM is also depends on material thickness. Utilizing this effect, can permittivity and thickness of material be simultaneously measured ? Please give some comment.

2) In this paper, the effect of meta-material is not necessarily obvious. Only figure 7 is provided to show the advantage of using meta-material. Please show the electric and magnetic field at resonant frequency. The field interact with meta-material at resonant frequency ? Please give some explanation about the origin of high sensitivity.

Author Response

(The authors gave the same response as above.)

Reviewer 3 Report

    The authors proposed a 1-D photonic crystal consisting of alternating metamaterial/silicon layers for measuring the permittivity of non-magnetic materials. The target material is used as the defect layer. The theoretical analysis on the transmittance is based on the transfer matrix method. The analytical results show the advantages (wide bandgap, sharp resonance, few layers needed) in using metamaterial over dielectric materials. Although the proposed design makes a meaningful contribution in the applications of photonic bandgaps, a few points need to be responded from authors before this manuscript can be accepted for publication.

(1) The authors need to address the advantage in permittivity measurement by using the proposed technique over other techniques (including the conventional methods).

(2) The values in permittivity and permeability of the metamaterial (equations (1) and (2)) were taken from the model proposed by Li, et al, in which the width of the structure is 3.8mm. However, the thickness of metamaterial layer of the current design is 31.5mm. Are the values in equations (1) and (2) still valid?

(3) Table 2 can be combined into Table 3. In addition to permittivity, also add a column of the permittivity calculated from equation (11) into Table 3 for comparison, and examine the accuracy of equation (11).

(4) The quality factor is defined in equation (8). However, there is not any data of Q reported.

(5) It is suggested that the transmittance of perfect crystal is added to Figure 2.

(6) In Figure 7, there is no mention of the permittivity of the defect layer (supposed to be 4).

(7) The format of reference is not consistent.

(8-1) The writing needs to be polished, for example, “The metamaterial is consists metallic fork between two metal split ring resonators. Such configuration have unique properties, such as a double negative refractive index in a certain range of frequencies. The metamaterial structure was simulated by Zhou Lei Li, et al. using finite difference time domain (FDTD) and obtained the effective electric permittivity ?? and effective magnetic permeability ?? were obtained [37], “.

(8-2) There are many short paragraphs beginning with “Where ..”

(9) The asterisk (*) for correspondence should be placed on the last author instead of the first author.

Author Response

(The authors gave the same response as above.)
